# Pharmacological Targeting of BET Bromodomain Proteins in Acute Myeloid Leukemia and Malignant Lymphomas: From Molecular Characterization to Clinical Applications

**DOI:** 10.3390/cancers11101483

**Published:** 2019-10-02

**Authors:** Diana Reyes-Garau, Marcelo L. Ribeiro, Gaël Roué

**Affiliations:** 1Laboratory of Experimental Hematology, Department of Hematology, Vall d’Hebron Institute of Oncology (VHIO), Vall d’Hebron University Hospital, Autonomous University of Barcelona, 08035 Barcelona, Spain; dreyes@vhio.net (D.R.-G.); mlribeiro@vhio.net (M.L.R.); 2Laboratory of Immunopharmacology and Molecular Biology, Sao Francisco University Medical School, Braganca Paulista, São Paulo 12916-900, Brazil

**Keywords:** bromodomain and extra-terminal domain, BRD2, BRD4, super-enhancer, NF-κB, MYC, combination therapy, hematological malignancies, protein degraders

## Abstract

Alterations in protein-protein and DNA-protein interactions and abnormal chromatin remodeling are a major cause of uncontrolled gene transcription and constitutive activation of critical signaling pathways in cancer cells. Multiple epigenetic regulators are known to be deregulated in several hematologic neoplasms, by somatic mutation, amplification, or deletion, allowing the identification of specific epigenetic signatures, but at the same time providing new therapeutic opportunities. While these vulnerabilities have been traditionally addressed by hypomethylating agents or histone deacetylase inhibitors, pharmacological targeting of bromodomain-containing proteins has recently emerged as a promising approach in a number of lymphoid and myeloid malignancies. Indeed, preclinical and clinical studies highlight the relevance of targeting the bromodomain and extra-terminal (BET) family as an efficient strategy of target transcription irrespective of the presence of epigenetic mutations. Here we will summarize the main advances achieved in the last decade regarding the preclinical and clinical evaluation of BET bromodomain inhibitors in hematologic cancers, either as monotherapies or in combinations with standard and/or experimental agents. A mention will finally be given to the new concept of the protein degrader, and the perspective it holds for the design of bromodomain-based therapies.

## 1. Introduction

In eukaryotic cells, acetylation of histone lysine is considered to be the most active protein post-translational modification. Mechanistically, the acetylation impacts the transcription through the neutralization of the positive charge of lysine to disrupt electrostatic interactions between histones and DNA, which leads to a transcriptionally active open state of chromatin [1]. The lysine acetylation is regulated mainly by three classes of proteins, the “writers” (histone acetyltransferases and histone methyltransferase), the “readers” (bromodomain (BRD)-containing) proteins, and the “erasers” (histone deacetylases and histone demethylases) [2,3,4,5,6,7]. 

Structurally, BRD consist of a left-handed bundle of four alpha helices, with interhelical loops forming a hydrophobic binding pocket that participates in acetyl-lysine identification [4]. In the human genome, the acetyl-lysine-binding BRD are observed over 46 distinct proteins [4,8]. The BRD proteins include, among others, chromatin-modifying enzymes, methyltransferases, helicases, chromatin remodelers, transcriptional co-activators and mediators, and the bromodomain and extraterminal domain-containing (BET) protein family [8]. It is well known that BRD can bind to transcriptional sites and regulate the activity of chromatin remodeling complexes, controlling a wide range of cellular events [2]. BRD-containing proteins are also engaged in the regulation of several signaling cascades, and could be activated upon oncogenic rearrangements, granting them with a key function in the development/progression of several types of cancer, including hematological malignancies. Among these factors, BET proteins represent a novel class of epigenetic readers with the ability to bind acetylated lysine residues in histone and nonhistone proteins, conferring them the capacity to control whole genome activity. The relevance of these proteins in the transcriptional control of malignant hematopoiesis makes pharmacological inhibition of BET protein activity particularly relevant to address the vulnerability of hematological cancers to BRD inhibitors [9]. Accordingly, the idea to target BRD has generated a great interest and culminated in the development of bromodomain inhibitors in a wide range of entities, including acute leukemia, multiple myeloma (MM), and aggressive B-cell lymphoma [2].

## 2. The BET Protein Family 

The BET family comprises four members, namely, BRD2, BRD3, BRD4, and the germ cell-specific BRDT, which act as regulators of RNA transcription and cell cycle progression through the activation of RNA polymerase II (Pol II). A universal feature of these proteins is the presence of two conserved N-terminal bromodomains (BD1 and BD2) that specifically recognize and bind to acetylated lysine residues on histone tails and other nuclear proteins [4]. 

Among the BET proteins, BRD2 was the first characterized as a nuclear noncanonical protein kinase with a role in the transduction of mitogenic signal [10,11]. BRD2 binds to acetylated H3 and H4 histones, recruits transcription factors, transcriptional co-activators, and transcriptional co-repressors, activating the transcription [12,13,14]. According to Denis et al. [14], the transcription complexes containing BRD2 and the chromatin remodeling machines are (i) TATA binding factor-associated factors and Pol II, (ii) activated transcription factors E2F and DP-1, (iii) Mediator proteins, (iv) chromatin/histone modification enzymes (HDAC11, CBP, and p300), and (v) SWI/SNF remodeling complex components. Together with BRD3, these complexes are needed for a permissive Pol II transcription through acetylated nucleosomes [15]. Initially, it has been demonstrated that BRD2 regulates cell cycle through the recruitment of E2F1 and E2F2, which are key transcriptional regulators of S phase genes [10,14,16]. Together with the fact that BRD2 is physically present at the *CCNA* promoter at both the G1 and S phases of the cell cycle, these data confirm both the role of BRD2 as a scaffold that mediates access of transcriptional control proteins to chromatin, and the functional link between BRD2 and proliferation [16].

### BRD4: Biological Roles and Molecular Mechanisms of Action 

The best known member of BET family is BRD4, which shares 80% identity at the amino acid level with BRD2 [17]. BDR4 is a transcriptional and epigenetic regulator that has a crucial role during embryogenesis, controlling cell cycles and maintaining genome stability. The role of BRD4 as a transcriptional regulator was initially proposed due to its interaction with both (i) cyclin T1 and CDK9 which belong to the active form of positive transcription elongation factor b (P-TEFb), and (ii) Mediator complex, a 30 subunit coactivator complex that physically interacts with BRD4 and P-TEFb [18,19]. Additionally, Mediator and BRD4 stabilize each other’s occupancy over the genome, and both cooperate in recruiting P-TEFb [19,20,21]. The first attempt to characterize BRD4 identified it as a protein associated with G1-S cell cycle progression [22]. Mechanistically, it has been shown that BRD4 is recruited to the promoters of G1 genes where it binds to acetylated histones using both BRD modules. The BD2 domain identifies and interacts with cyclin T1, which is particularly important to maintain Pol II in the promoter region of active genes, leading to transcription initiation and elongation of a large set of genes related to cell growth, including *MYC* and its target genes [23,24,25,26]. ChIP-seq data have shown that BRD4 co-localizes at the nucleosome-free site occupied by transcription factors (TFs) at enhancers and promoters [27,28]. Furthermore, it was demonstrated that BRD4 also forms super enhancer complexes with the Mediator complex, favoring the association of transcription regulating proteins, regulating then the expression of some oncogenic drivers in a large set of cancers [29]. 

Beside these functions, BRD4 also has an important role in mediating inflammatory transcriptional cascades by interacting with acetylated nuclear factor kappa B (NF-κB) subunit RELA (also known as p65). Upon stimulation, RELA is acetylated at lysine 310 through the p300/CBP coactivators, which maximizes the transcriptional activation of NF-κB [30]. Subsequently, Huang et al. showed that acetylated RELA activates NF-κB through the recruitment of BRD4 via specific interaction between the acetylated lysine-310-BRD4 bromodomains. BRD4, then activates CDK9, which phosphorylates PolII, thus promoting NF-κB transcriptional signaling [31]. In parallel, BRD4 plays a structural role supporting the higher chromatin architecture [32]. Subsequently, Devaiah et al. showed that BRD4 can act as a histone acetyltransferase by acetylating H3K122 residue, leading to nucleosome destabilization and clearance accompanied by chromatin decompaction. Thus, an upregulation of BRD4 might lead to chromatin remodeling, followed by reduced nucleosome occupancy and increased gene transcription [33] (Figure 1). Beside its pivotal role in controlling cell cycles, BDR4 is also committed with nonhomologous end-joining (NHEJ) DNA repair [34,35]. In B lymphocyte biology, it has been reported that BDR4 is required during immunoglobulin isotype switching for the accomplishment of class switch recombination after DNA double strand breaks (DSBs) by Activation Induced cytidine Deaminase (AID) [34]. It is known that DNA DSBs are followed by H4 acetylation and γH2AX, which induces BRD4 recruitment. Amongst many DNA repair players that interact with BRD4, 53BP1 is its major binding partner in DNA damage regulation. The interplay of BRD4 at DSBs maintains the binding of 53BP1 with DNA repair complexes on site, promoting the NHEJ activity [34,35]. In addition, BRD4 has been also involved with the activation of DNA damage checkpoint signaling in a transcriptionally independent manner. In this sense, BRD4 interacts and regulates the function of pre-replication factor CDC6, which is essential for the activation of replication checkpoint response [36]. Recently, it has been highlighted that BRD4 has a nontranscriptional role controlling telomere homeostasis. Both the treatment with BET inhibitors and BRD4 knock-down lead to the downregulation of telomerase reverse transcriptase (TERT) and an impairment of telomerase activity, followed by a decrease in the recruitment of histone active marks [37]. Similarly, Wang et al. showed that a long-term treatment of mouse and human cells with BDR4 inhibitors led to a telomere reduction, however without affecting the telomerase enzymatic activity [38]. Although BRD4 has an important role in telomere maintenance, the mechanism(s) by which it occurs is not fully understood. It has been hypothesized that BRD4 might serve as a platform, recruiting and stabilizing the binding of the telomerase complex, leading to telomeres lengthening [29].

## 3. The Role of BET Proteins in Hematological Cancers

As mentioned previously, the BET family of proteins are important epigenetic regulators involved in promoting gene expression of critical oncogenes by keeping an abnormal chromatin state in various hematologic malignancies, including MM, acute myelogenous leukemia (AML), acute lymphoblastic leukemia (ALL), diffuse large B-cell lymphoma (DLBCL), Burkitt lymphoma (BL), and mantle cell lymphoma (MCL) [39,40,41,42]. Among the oncogenes known to be regulated by BET proteins in these diseases, *MYC* is probably the most relevant, as it is overexpressed in about 60%–70% of all cancers [43].

### 3.1. MYC-Driven Mechanisms of Oncogenesis Regulated by BET Proteins

Mechanistically, MYC accumulates into promoter regions acting as a transcriptional amplifier overreacting gene expression profile [44]. MYC recruitment increases in histone lysine acetylation sites which is associated with transcriptional activation [45]. This oncogene also enhances transcription elongation through the recruitment of P-TEFb and the Mediator complex [25,46]. 

In hematopoietic malignancies, BRD4 was found to exert a wide role in keeping MYC stable expression, providing a rationale for the use of BET inhibitors for the targeting MYC-dependent transcription [39,41,42]. In B-cell lymphomas it has been reported that chromosomal translocations involving *MYC* locus are common to germinal center B-cell (GCB) DLBCL, with about 32% of DLBCL patients harboring MYC overexpression [47]. Likewise, BL is also characterized by translocations involving the fusion of an immunoglobulin heavy- or light-chain gene promoter to *MYC* locus, which leads to MYC expression in the B-cell compartment and consequent lymphomagenesis [48]. Given the relevance of MYC in DLBCL and BL, BET inhibitors are currently being tested in clinics (see below). Additionally, it has been observed that BRD4 also plays a crucial role in AML maintenance through MYC activation and aberrant transcriptional elongation, the suppression of BRD4 being a major cause of cell-cycle arrest and apoptosis induction [39].

### 3.2. BET-Mediated Expression Regulation of Other Oncogenes

In addition to MYC, BRD4 also regulates the expression of several regulators of the cell cycle, including CDK4/6 and cyclin D1 [49,50]. These findings are particularly relevant in MCL, as this entity is characterized by the t(11;14) translocation that fuses the *CCND1* gene, coding for cyclin D1, to the immunoglobulin heavy locus gene, leading to the overexpression of the cyclin, thus increasing CDK4/6 activity with the consequent deregulation of the cell cycle at the G1 to S transition [51]. 

Another relevant driver gene interacting with bromodomain proteins is nucleophosmin (*NPM1*). Epidemiological data have shown that about 35% of AML cases harbor *NPM1* mutation, this alteration being one of the most common distinctions in AML [52]. Different functions have been attributed to NPM1, which include (i) ribosomal assembly, as a nucleolar histone chaperone, and (ii) the regulation of the ARF-p53 pathway [52]. Such a role depends on drive back and forth between the nucleus and cytoplasm. Functionally, NPM1 interacts with BRD4 in the nucleus and HEXIM1 represses BRD4-mediated transcriptional elongation [53]. So, considering that the NPM1 loss of function mutation leads to a cytoplasmic localization of the NPM1 [54], transcriptional repression of BRD4 by NPM1 is impaired, leading an increased expression of MYC and BCL2 [53]. Recently, BRD4 has also been described as an important regulator of AML cell autophagy through distinct complementary mechanisms including (i) a direct modulation of autophagy-related genes, and (ii) an indirect increase of reactive oxygen species release by KEAP1 (kelch like ECH associated protein 1), followed by a blockade of the NRF2 (nuclear factor, erythroid 2 like 2) antioxidant pathway [55].

Chromosomal translocations involving the *lysine methyltransferase 2A* (*KMT2A*) gene occur in AML patients, being rare in adults (5% of cases) and more common in children (22% of cases) [56]. This phenomenon is also observed in approximately 10% of ALL cases [57]. *KMT2A* encodes for a histone methyltransferase involved in transcriptional regulation [57,58]. *KMT2A* translocation leads to a fusion of KMT2A with different regulators of transcriptional elongation such as the super elongation complex (SEC), which leads to a deregulated transcription driving leukemogenesis. BRD3 and BRD4 are components of the SEC and the polymerase-associated factor complex, which plays a role in KMT2A localization and transcription [59]. Hence, NPM1-mutated AML cases, as well as patients harboring translocations involving *KMT2A* gene, represent a suitable target population for BET inhibitor (BETi) therapy. 

## 4. Characterization and Evaluation of BET Inhibitors

### 4.1. Preclinical Activity of BET Inhibitors as Monotherapies in Hematological Malignancies 

Small molecules capable of disrupting the BET protein’s interaction with acetylated histones were first characterized from chemical compounds targeting the acetyl lysine binding pocket of the histone acetyltransferase p300/CREB-binding protein (CBP), thus preventing its association with the tumor suppressor protein p53 during *CDKN1A* transcription in response to DNA damage. These molecules were found by nuclear magnetic resonance (NMR) spectroscopy screening from a library of compounds selected according to their capacity to disrupt this specific protein–protein association [60]. 

Subsequently, several BET inhibitors with an improved affinity for the BRDs have been developed and tested in experimental studies in order to get distinct therapeutic approaches for the treatment of a wide range of malignancies. Among these molecules, the thieno-triazolo-1,4-diazepine JQ1 is a competitive interactor of the BRD2 and BRD4 acetyl-lysine binding motifs [42,61]. The antiproliferative activity of this compound was first characterized in a panel of human leukemia and lymphoma cell lines, achieving a 50% inhibition of cell proliferation in almost all the cell lines tested, at concentrations ranging between 50 nM and 500 nM [42]. Mechanistically, this compound was described to downregulate *MYC* expression by promoting BRD2 and BRD4 displacement from *MYC* promoter region in both human lymphoma and T-ALL cell lines [42,62], as well as in mouse models of MM [41]. JQ1 was also described to directly target BRD4-Nuclear Protein in Testis (NUT) oncoprotein interaction [63]. This property grants the drug with the capacity to abrogate cell proliferation in vitro and to inhibit tumor growth in mice xenografted with a panel of cell lines bearing different *BRD4*–NUT translocations, thus suggesting a therapeutic potential for the treatment of NUT midline carcinoma (NMC) [61].

These promising results led to the design and development of other BET inhibitors structurally related to JQ1. Among them, CPI203 showed an improved bioavailability when compared to its precursor [64,65] together with a remarkable antitumoral activity in different preclinical models of B-cell non-Hodgkin lymphomas (B-NHLs), including MCL [66], DLBCL [67,68], MM [69], and MYC+/BCL2+ double hit lymphoma [70]. Originated from the same company, the benzoisoxazoloazepine CPI0610 was assessed in in vivo studies in mice, where its tumor reducing activity appeared to correlate with a BET-driven reduction in *MYC* gene expression. Moreover, CPI0610 treatment also impaired the production of pro-inflammatory cytokines through its action over the NF-κB pathway [71], and the compound displayed potent cytotoxicity against MM cell lines and primary cultures xenografted in mice by triggering G_1_ cell cycle arrest and caspase-dependent apoptosis, even in the presence of protective stroma [72]. 

Later on, two other BET inhibitors were developed by Gilead Sciences Inc., namely GS-5829 and GS-626510, which demonstrated a good capacity to bind reversibly BRD2, BRD3, and BRD4 BET proteins, thus preventing the recruitment of these latest to acetylated histones [73]. GS-626510 has also been shown to regulate the production of a set of cytokines when combined with the PI3Kδ inhibitor, idelalisib, in cocultures involving M2-polarized macrophages and DLBCL primary samples [74]. 

The thienotriazolodiazepine compound MK-8628/OTX015 (birabresib) has also been demonstrated to efficiently prevent the binding of BRD2, BRD3, and BRD4 to acetylated histone H4. Several works have pointed out its capacity to promote cell cycle arrest, together with the downregulation of MYC, BRD2, and BRD4 protein levels and upregulation of the PolII negative transcription regulator, HEXIM1, in human AML and ALL cell lines and primary samples [75]. Additionally, this compound has been reported to induce apoptosis in non-GCB subtypes of DLBCL. In agreement, MK-8628/OTX015 has been demonstrated to alter the *MYC* and E2F1-dependent gene expression signature by downregulating the expression of signaling proteins belonging to the NF-κB, Toll-like receptor (TLR), and Janus kinase/signal transducers and activators of transcription (JAK/STAT) pathways, including *IRAK1*, *TLR6*, *TNFRSF17, MYD88*, *IRAK1*, *IL6*, and *IRF4* [76]. Additionally, ATAC-Seq analysis performed with the SET-2 AML cell line treated with this compound revealed its capacity to impair the chromatin accessibility of transcription initiation complexes at genome wide level [77]. 

The dihydroquinazolinone PFI-1 (PF-6405761), initially designed as a BET chemical probe, exhibited activity against leukemic cell lines carrying *MLL* rearrangements by impairing their clonogenic growth upon cell cycle arrest at the G_1_ phase and inducing apoptosis, with a concomitant decrease in the expression of MYC and one of its target genes, *AURKB*, which promoted a global reduction in Histone H3 phosphorylation at Ser10 [78].

Similarly to PFI-1, the benzodiazepine I-BET762 also binds the BRD2 and BRD4 acetyl-lysine binding pockets and prevents their interaction with the acetylated histone tails. This molecule has shown remarkable apoptogenic activity in human and mouse lymphoma cells, mediated by a p53-independent and BCL-2 family-dependent activation of the intrinsic mitochondrial apoptotic pathway [79]. 

Among the BET inhibitors with prominent pro-apoptotic activity, the BRD2/BRD4-selective BETi ABBV-075 strongly triggered apoptosis in AML, NHL, and MM cell lines, and had mainly a broad antiproliferative activity across different solid tumor-derived cell lines [80]. Formulations containing ABBV-075 are currently in clinical trials for the treatment of advanced hematologic malignancies (see below).

Importantly, the pan-BET inhibitor bromosporine (BSP), with a nanomolar affinity for 13 bromodomains and low micromolar affinity for 12 additional ones (either BET or non-BET BRDs-containing proteins), has been developed to determine the role of BET proteins in leukemic cell lines. BSP showed similar effects to JQ1 towards the modulation of cell proliferation and clonogenic growth, whereas it did not show almost any of the characterized gene expression effects when specifically inhibiting non-BET proteins, thus indicating that only BET proteins (and not other BRDs-containing factors) have a predominant role in leukemogenesis [81].

Recently, a novel BETi named INCB054329 has been evaluated both in preclinical and phase I clinical trials. The chemical structure of this compound was designed to promote its interaction with both BRD4 BD1 acetyl lysine binding pockets and tryptophan-proline-phenylalanine (WPF) shelf. In preclinical works, INCB054329 demonstrated significant antiproliferative effects over 32 cell lines derived from hematological malignancies including AML, NHLs, and MMs, as well as a significant reduction of tumor growth for OPM-2 myeloma cell lines xenografted in mice. In particular for myeloma cells, INCB054329 also evidenced pro-apoptotic properties. Both antiproliferative and pro-apoptotic effects were driven by a mechanism involving BRD4 displacement from MYC, FGFR3, IL6R, and NSD2 enhancers, with a concomitant reduction in the corresponding mRNA levels and an increase in myeloma cells sensitivity to JAK inhibitors, thus arising the possibility of obtaining therapeutic advantages from the combination of these two kind of compounds [82]. The mechanisms of action of BET inhibitors are summarized in Figure 2.

### 4.2. Evaluation of BET Inhibitors as Part of Combination Therapies in Preclinical Settings

Given their high activity against a number of signaling pathways crucial for the promotion of lymphoid tumors, BET inhibitors have also been evaluated in numerous combination trials in different subtypes of hematological malignancies. 

Among the tested inhibitors, JQ1 was used in vitro and in mouse xenografts of activated B-cell (ABC) DLBCL subtypes in combination with ibrutinib, the first-in-class inhibitor of Bruton’s kinase (BTK). The rationale behind the simultaneous use of those agents relies on the fact that both BCR and BRD2/BRD4 activity promote the activation of IKK, an apical kinase of the NF-κB pathway known to be over-activated in ABC-DLBCL. Accordingly, JQ1 has been demonstrated to potentiate ibrutinib activity in both in vitro and in vivo models of ABC-DLBCL, mediated by synergistic inhibition of NF-κB-driven signaling [68].

Synergistic antiproliferative effects were also observed with OTX015 when tested over a panel of five DLBCL cell lines in combination with either mTOR, PI3K-delta, or BTK inhibitors (everolimus, idelalisib, and ibutrinib respectively), immunomodulatory drugs (IMiDs, lenanidomide), DNA methyltransferase inhibitor (decitabine), chemotherapy agents (doxorubicin and bendamustine), or the anti-CD20 targeting antibody rituximab [76]. In line with those findings, antitumoral effects have also been described for OTX015 when assessed over SUDHL-2-derived mouse xenografts in combination with different biological agents including the HDACi vorinostat, everolimus, rituximab, or ibutrinib [83].

Considering the role of *MYC* oncogenic translocation described in BL and [84,85], many efforts have been put in the identification of synergic combinations between different BET inhibitors and other biological agents. In this direction, Tomska et al. reported the results of a high throughput combinatorial screening of different inhibitory compounds, and their effect on the cell viability of 32 different blood cancer cell lines, including 17 BL lines. Despite the high heterogenic results attributed to the large subset of cell lines, the authors pointed out the strong synergistic effects of OTX015 in combination with the CDK2/7/9 inhibitor SNS-032 and the PI3K/AKT/mTOR pathway inhibitors idelalisib, duvelisib, MK-2206, everolimus, and deforolimus [86].

Other combinatorial treatments assessed in preclinical settings included JQ1, CPI0610, ABBV-075, MK-8628/OTX015, and I-BET762 in association with and the HDAC inhibitors vorinostat and romidepsin, or the BCL-2 inhibitor venetoclax, in preclinical models of cutaneous T-cell lymphoma (CTCL) including patient-derived primary samples and CTCL cell lines [87]. The rationale behind these different combinations relied (i) on the fact that, whereas HDACs regulate acetylation levels of DNA-associated proteins [88], BET proteins bind acetylated histones, promoting initiation of transcription and (ii) on the demonstration that transcription of the *BCL2* gene was impaired upon BET protein inhibition [89,90,91]. Accordingly, authors observed additive effects in the inhibition of the cell cycle regulators MYC, NF-kB, cyclin D1, and IL5Rα in the promotion of the expression of the negative cell cycle regulator CDKN1A, and in the induction of some death receptor/ligand family members (TRAIL, FasL, DS4, DS5 and TNFR1) upon simultaneous exposure of human CTCL cell lines to BET inhibitors and HDAC inhibitors [92]. 

The BETi ABBV-075 has been evaluated as a putative agent to overcome resistances to venetoclax-mediated BCL-2 inhibition in both AML cell lines as well as in models of AML blast progenitor cells (BPCs) engrafted in immune-compromised mice. The rationale underlying the efficacy of this combination was that single agent treatment with venetoclax induced the expression of the anti-apoptotic MCL-1 as a compensatory mechanism, whereas ABBV-075 was able to remarkably block the transcription of this BRD4-controlled gene [93]. 

Following the previous attempts to combine BET inhibitors and epigenetic drugs, other combinations, including the BETi RVX2135 and the HDAC inhibitors vorinostat or LBH-589, have also been assessed in MYC-induced murine lymphoma cells, showing a synergistic pro-apoptotic effect, which was believed to rely on BET inhibitors capacity to induce the re-expression of several HDAC-silenced genes [94].

Likewise, other combinatorial treatments successfully tested in preclinical trials for DLBCL include the BETi CPI203 and the CXCR4 inhibitor IQS-01.01RS. The rationale of this combination relies on the malignant cells’ capacity to overexpress CXCR4, allowing them to interact with the corresponding stromal ligand, the chemokine CXL12, and to colonize the vascular and endosteal niches, increasing the probabilities of relapse after treatment with (chemo)-immunotherapy. Overactivation of the CXCR4-CXL12 axis leads also to hyperphosphorylation of ERK and AKT, and its inhibition with the CXCR4 antagonist led to a reduction in the phosphorylation levels of these proteins and to the destabilization of MYC protein. Accordingly, IQS-01.01RS and CPI203 combinatorial treatment achieved a synergistic downregulation of MYC an improved tumor growth inhibition in DLBCL cell lines and xenograft model [67]. Thanks to its capacity to modulate BCL-2 and BCL-2-like protein levels, and to block IRF4/MYC signaling, CPI203 also exerted notable activity either as single agent or in combination with the BCL-2 antagonist venetoclax in *MYC+/BCL2+* DHL [70], or combined with the proteasome inhibitor bortezomib or lenalidomide in bortezomib-resistant MCL [66].

Of special interest, a combination of CPI0610 with the IMiDs lenalidomide and pomalidomide showed also in vitro synergism, involving the inhibition of the IKZF1/IRF4/MYC signaling axis [72] and warranting the clinical evaluation of this combination strategy in MM patients. 

Similarly JQ-1, IBET151 and PFI-1 have also been shown to synergistically interact with the IMiDs in in vitro and in vivo models of primary effusion lymphoma (PEL) [95]. 

With regard to the INCB054329 inhibitor and its potential to downregulate IL6R expression and sensitize myeloma cells to JAK2 inhibitors, preclinical studies combining this compound with inhibitors of the JAK-STAT pathway such as ruxolitinib or itacitinib demonstrated synergistic effects and enhanced efficiency in reducing myeloma cell proliferation and MM tumor growth [82].

Of note, in almost all the above-cited experimental studies, it has been suggested that beside the cytostatic activity of the different BET inhibitors when used as single agents, additive and/or synergistic effects with drugs targeting distinct regulatory proteins underlined an enhanced cytotoxicity and/or capacity to overcome resistances to standard or experimental agents, thus warranting their evaluation in clinical settings in the majority of cases [96].

## 5. Targeting BET in Hematological Cancer Patients

The safety, tolerability, pharmacokinetics, pharmacodynamics, and clinical activity of BET inhibitors have been extensively tested in clinical trials for distinct cancers, including hematological malignancies (Table 1). The OTX015 compound was the first one that underwent clinical trials for the treatment of AML. Although no correlation was found between the degree of response and the presence of somatic mutations in 42 genes related to AML including *NPM1*, *IDH2*, *FLT3*, *EV11*, and *KMT2A*, preliminary activity was observed in five of the total 41 enrolled patients (NCT01713582) [97]. OTX015 was also evaluated in a cohort of 45 patients with lymphoma (*n* = 33) and MM (*n* = 12), with three DLBCL patients achieving complete remission lasting between 4 and 13 months, although no association between MYC expression levels and OTX015 sensitivity could be identified (NCT01713582) [97]. Regarding treatment-related toxicity, although the authors showed that thrombocytopenia is an early adverse event during OTX015 treatment, this condition was manageable at a dose of 80 mg, once a day on 14 days-on 7 days-off schedule. Moreover, doses above this level were associated with increased toxicity without a clear effect on efficacy. The most common nonhematological toxicities observed were gastrointestinal and cutaneous events, fatigue, metabolic disorders, and hyperglycaemia [97]. The reported adverse effects for each BET inhibitor tested in clinical trials are summarized in Table 2. 

Further clinical trials were also carried out with either OTX015 or INCB054329, including an interventional dose exploration study with OTX015 involving AML and DLBCL patients, or an open-label dose-escalation study with INCB054329 for the treatment of advanced solid tumors and hematologic malignancies. Unfortunately, whereas the first one had to be discontinued due to limited efficacy, the second one was withdrawn due to the high pharmacokinetics (PK) variability associated to the compound (NCT02698189, NCT02431260) [98].

Regarding CPI0610, a number of clinical trials have been launched for the treatment of certain hematological malignancies including MM (NCT02157636), AML, myelodysplasic/myeloproliferative syndrome (NCT02158858), myelofibrosis [99], and progressive ABC-DLBCL (NCT01949883). Data emerging from these clinical trials indicates that CPI0610 is well tolerated and has activity mainly in patients with relapsed/refractory lymphoma [100]. Given the role of BET proteins in the control of different processes associated to myeloproliferative disorders such as production of inflammatory cytokine, platelet counts, spleen volume, or bone marrow fibrosis [74], CPI0610 is also currently being evaluated in a phase II study as a monotherapy and in combination with the JAK2 inhibitor ruxolitinib in patients with myelofibrosis who are not eligible to receive a JAK inhibitor or have had an inadequate response to ruxolitinib [75].

GS-5829 is another BET currently being evaluated in clinical for its efficiency and tolerability to treat advanced solid tumors and hematological cancers, thanks to the demonstration of its safety profile and efficacy in rodents (NCT02392611).

As discussed above, BET inhibitors have been successfully used in combination in experimental studies, showing synergistic effects when combined with other anticancer drugs. These data suggest that rather than monotherapy, the use of BETi-based combinational therapy with other antitumoral agents might play an important role in the future. Indeed, there are several ongoing clinical trials currently evaluating BET inhibitors in combination with azacitidine, daratumumab, entinostat, exemestane, fulvestrant, molibresib, rituximab, ruxolitinib, and venetoclax for the treatment of hematological malignancies (Table 1 and Table 2).

## 6. Mechanisms of Resistance to BET Inhibitors 

As previously mentioned, constitutive activation of the BRD4-MYC axis is a general hallmark of hematological disorders, and in these models the use of BET inhibitors circumvents the MYC transcriptional program, leading to a consistent antitumor effect [49]. Although initial results generated from both preclinical and clinical evaluation of the different BET inhibitors are in general promising, the antitumoral capacity varies greatly among these agents [42]. Furthermore, recent research has shown that cancer cells can acquire resistance to this class of drug after prolonged treatment. To date, the described mechanisms of acquired resistance to BET inhibitors have not been related to the acquisition of somatic mutations or changes in copy number of the BET bromodomain genes, but rather to secondary adaptation consisting on compensatory changes in the activation of mechanisms that regulate the initiation and elongation of transcription. Some works have reported these kind of adaptive response in AML cells, in which the acquisition of resistance to BET inhibitors has been linked to a downmodulation of BRD4 protein and activation of the NF-κB pathway without altering the expression levels of MYC, coinciding with an aberrant activation of the WNT/β-catenin and the TGF-β signaling pathways [101].

A mechanism of resistance to BET inhibitors in AML cells suggested by Rather et al. consists of the acquisition of BRD4 independence to drive *MYC* transcription trough the suppression of PRC2 complex activity [102]. Other works carried out in triple negative breast cancer cells (TNBCs) have shown the acquisition of resistance to be associated with the hyperphosphorylation state of BRD4 as a consequence of a reduction in the expression of the BRD4 phosphatase, PP2A, and a concomitant enrichment in the levels of the transcriptional activator associated with BRD4, MED1 [103].

Another mechanism of escape to BET inhibitor’s activity is a defective intrinsic mitochondrial cell death, as described in the *Eµ*-*myc* transgenic mouse model and in human B-cell lymphoma samples. In this work, short-term resistance to I-BET762 is attributed to the deletion of apoptotic peptidase initiating factor-1 (APAF-1), a key transducer of the apoptotic cascade elicited by mitochondrial apoptogenic factors. Nevertheless, this resistance can be overcome after a compensatory mechanism involving the induction of autophagy as suggested by an increase in the active form of LC3, a hallmark of autophagy, when depleting the cells from APAF-1. Interestingly, long-term resistance to I-BET762-mediated apoptosis had been described as a result of induction of anti-apoptotic BCL-2 [79].

## 7. Future Generations of Bromodomain Inhibitors

Over recent years, a different and novel approach to target BETs has been developed in order to inhibit this family of proteins, through specific BET degradation mediated by the use of Proteolysis-Targeting Chimaeras (PROTACs). To date, this approach has been only used to target the BRD4 protein. Functionally these molecules arise from the fusion between a hetero-bifunctional synthetic ligand composed of the corresponding targeted BET bromodomain’s ligand (which functions as a PROTAC anchor), and the von Hippel–Lindau (VHL) E3 ubiquitin ligase, which triggers BET protein ubiquitination and further proteasome-mediated degradation [104,105]. According to the BRD’s selectivity, three BET-PROTACs have been developed to date that can target both BRD2 and BRD4 degradation—dBET1, ARV-825, and MZ1, this latest being the one with the highest potency [106]. As a functional validation, these molecules efficiently induced apoptosis in MCL-derived cells resistant to ibrutinib, and allowed prolonged survival of MCL xenografts, an effect that was superior to that observed in OTX015-treated animals [107]. On the other hand, dBET1 was shown to evoke a rapid degradation of BRD4, followed by the blockade of MYC transcription, and induction of apoptosis in AML cell lines and primary cultures, as well as in an AML xenograft model [108]. Based on a similar synthetic scheme, the optimized small-molecule degrader dBET6 allowed extending observations on bromodomain-independent functions in T-ALL. In this model, acute loss of BRD4 elicited a global disruption of productive transcription elongation and a collapse of the core regulatory circuitry, more resembling CDK9 inhibition than BET bromodomain displacement, and thus redefining the role of BRD4 in global gene regulation [109].

Another facet of BRD4 that has been exploited for therapeutic purposes is its atypical kinase activity [110], which led to the screening of a panel of kinase inhibitors with the ability to bind Ac-K binding pockets of BRD4. This study led to the identification of PLK1- and JAK2-interfering drugs as potent and selective inhibitors of BRD4-BET kinase activity [111]. Although promising results have been reported for this new generation of BET-targeting agents in preclinical trials, their therapeutic window when moving to clinical trials has still to be evaluated.

## 8. Conclusions

BET protein inhibitors are the first example of a successful pharmacological blockade of epigenetic readers, and are a large and diverse group of epigenetic regulators that recently gained interest in several hematological cancers. Unlike previous epigenetic drugs, an in-depth mechanistic characterization preceded the clinical evaluation of this new class of agents in different settings, including refractory AML, MM, and aggressive B-cell lymphoma. The first reports of clinical activity in these highly treated patients with acceptable off-target effects are supporting further progress to phase II trials. Along with the insights into the alterations of bromodomain-containing proteins in the biology of hematologic malignancies, the realm of drug discovery in this field is expanding fast, with emerging new concepts and technologies for targeted molecular drug development. Newer approaches including BET protein degradation and BET-kinase inhibitors are now changing the landscape of available targeted therapies. Future research regarding these new classes of agent will include the identification of biomarkers and the evaluation of different kinds of combination therapies.

## Figures and Tables

**Figure 1 cancers-11-01483-f001:**
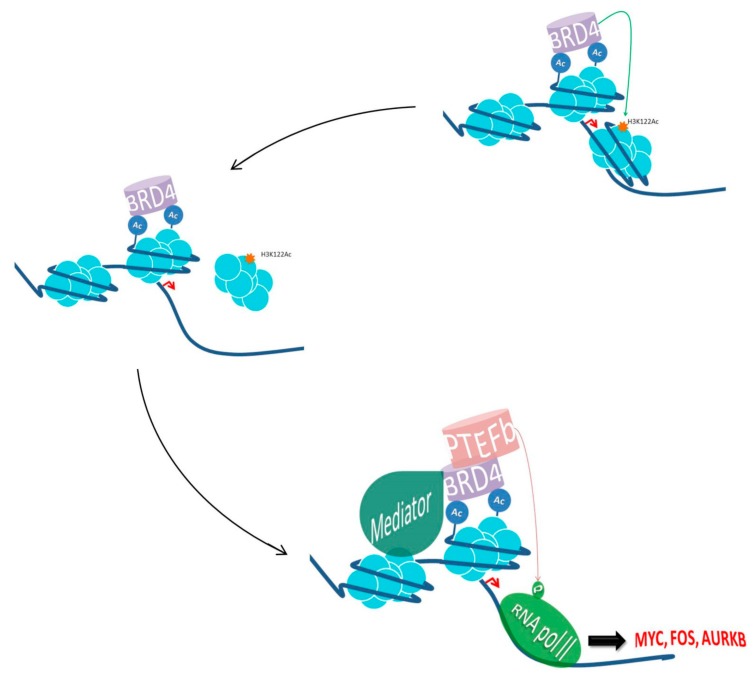
**Schematic representation of BRD4 transcriptional activation.** BRD4 acetylates histone tail lysines (Ac), and eventually H3K122, leading to nucleosome clearance and the recruitment of Mediator complex favoring and stabilizing the binding of RNA-PolII. BRD4 also interacts and activates P-TEFb, stimulating RNA-PolII to promote transcriptional activation of factors involved in cell cycle control and proliferation such as Aurora B, Cyclin D, E2F, MYC, FOSL1, TCF4 and Wnt5a.

**Figure 2 cancers-11-01483-f002:**
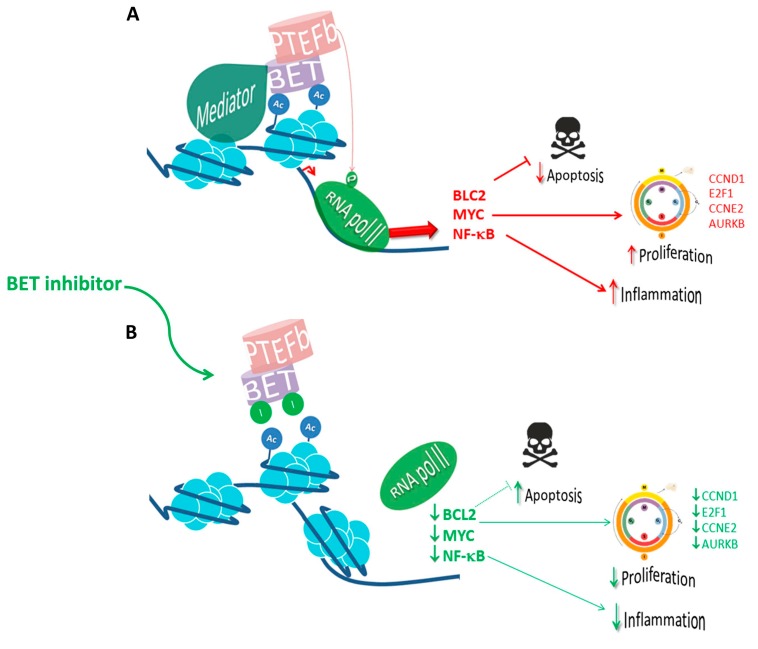
**The mechanism of action of BET inhibitors in the clinics**. (**A**) BET proteins regulate the transcription of genes related to multiple functions, including cell cycle, proliferation, and inflammation. The BET family member BRD4 has an important role in keeping oncogenic expression of MYC in hematopoietic malignancies. In addition, BRD4 also regulates BCL-2 leading to a decrease in apoptosis. Furthermore, it induces cell proliferation by upregulating (↑) CCND1, E2F1, CCNE2, and AURKB and is responsible for the transcriptional activation of NF-κB inflammatory response. (**B**) BET inhibitors compete with acetylated residues releasing BRDs from chromatin, reducing RNA-Pol II blocking (↓) transcription of downstream genes. The red arrows (→) indicate the oncogenic pathways mediated by BET, the green arrows (→) highlight the effect of BET inhibitors.

**Table 1 cancers-11-01483-t001:** Ongoing clinical trials evaluating BET inhibitors in hematological cancers.

BET Inhibitor	Target	Drug Combination	Disease	Study Phase (Number)	Status
**GSK525762/C**	BRD2/3/4	-	MM	I (NCT01587703)	Active, not recruiting
**OTX015/MK-8628**		Entinostat, Molibresib	Lymphomas	I (NCT03925428)	Not yet recruiting
		-	AML, MDS, DLBCL	I (NCT02698189)	Active, not recruiting
**CC-90010**	BDR2	-	R/R-NHL	I (NCT03220347)	Recruiting
**CPI0610**		Ruxolitinib	AML, MDS	II (NCT02158858)	Recruiting
**ABBV-744**	BRD4	-	AML	I (NCT03360006)	Recruiting
**BI 894999**		-	DLBCL	I (NCT02516553)	Recruiting
**BMS-986158**		-	Lymphoma	I (NCT03936465)	Recruiting
**RO6870810**		Daratumumab	MM	I (NCT03068351)	Active, not recruiting
		Venetoclax, Rituximab	R/R-DLBCL, High-grade B-cell lymphoma	I (NCT03255096)	Active, not recruiting

Abbreviations: Acute lymphoblastic leukemia (ALL), Acute myeloid leukemia (AML), Burkitt lymphoma (BL), Diffuse large B-cell lymphoma (DLBCL), Multiple myeloma (MM), Myelodysplastic syndromes (MDS), Non-Hodgkin lymphoma (NHL).

**Table 2 cancers-11-01483-t002:** Efficacy and toxicity of BET inhibitors in patients with hematological cancers.

BETi	Drug Combination	Target Disease and Criteria of Inclusion	Clinical Trial (Number of Patients, Age)	Response	Adverse Effects
**ABBV-075**	Venetoclax	**AML, MM**	NCT02391480 (*n* = 128, ≥18 years old)	No data available	No data available
**FT-1101**	Azacitidine	**R/R AML** (FLT3-ITD or FLT3-TKD mutated) **MDS** (eligible to receive azacitidine), **R/R NHL** (primary mediastinal, DLBCL and B-cell lymphoma)	NCT02543879 (*n* = 94, ≥18 years old)	No data available	No data available
**OTX015/MK-8628**	-	**AML** **Ph + ALL** **MM** (exposed to at least one alkylating agent/corticosteroid/IMiD and bortezomib) **DLBCL** (≥1 nonirradiated tumor mass ≥750 mm^3^). (failure of all standard therapies and life expectancy ≥3 months)	NCT01713582 (*n* = 141, ≥18 years old)	DOR in 6,6% of DLBCL patients and clinical activity without ORC in 13,3% patients	**Serious adverse effects:** Thrombocytopenia, febrile neutropenia, leukocytosis, sinus bradycardia, gastrointestinal events (grade 1–2), fatigue, asthenia (grade 1–2) and infections
	Azacitidine	**AML**	NCT02303782 ^†^	No data available	No data available
**CPI0610**	-	**MM** (progressed after at least one line of standard therapy)	NCT02157636 (*n* = 30, ≥18 years old)	No data available	No data available
	-	**Lymphoma** (progressed after/prior treatment and without available/effective standard therapy)	NCT01949883 (*n* = 64, ≥18 years old)	No data available	No data available
**PLX51107**	-	**R/R AML** or **NHL** (life expectancy ≥3 months)	NCT02683395 ^††^ (*n* = 50, ≥18 years old)	No data available	No data available
**BAY1238097**	-	**MM** (refractory to standard treatment or with no standard therapy available, life expectancy ≥3 months)	NCT02369029 ^††^ (*n* = 8, ≥18 years old)	No data available	No data available
**GS-5829**	Exemestane, Fulvestrant	**DLBCL, PTCL** (refractory to or intolerant of standard therapy or no standard therapy available)	NCT02392611 (*n* = 33, ≥18 years old)	No data available	No data available
**INCB054329**	-	**DLBCL, AML, MM, ALL, BL** (progressed following at least 1 line of therapy without further approved therapy available)	NCT02431260 ^††^ (*n* = 69, ≥18 years old)	PK variations among individuals	**Serious adverse effects** Anemia, neutropenia, thrombocytopenia, gastrointestinal disorders, fatigue, hyperbilirubinemia, pneumonia, multi-organ failure, hepatic vein thrombosis, infections increased alanine and aspartate aminotransferases, hyperbilinubinemia, respiratory, thoracic, and mediastinal disorders
**RO6870810**	-	**R/R AML** and **MDS** (allogeneic stem cell transplant not treated with immunosuppressors for ≥2 weeks, life expectancy ≥2 months)	NCT02308761 (*n* = 26, ≥18 years old)	No data available	No data available

Abbreviations: Acute lymphoblastic leukemia (ALL), Acute myeloid leukemia (AML), Burkitt lymphoma (BL), Diffuse large B-cell lymphoma (DLBCL), Multiple myeloma (MM), Myelodysplastic syndrome (MDS), Non-Hodgkin lymphoma (NHL), Relapsed/Refractory (R/R), Durable Objective Response (DOR), Objective Response Criteria (ORC). ^†^ Withdrawn ^††^ Terminated. Philadelphia chromosome positive (Ph+), Immunomodulatory drug (IMiD).

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
