# Peer review of "Pharmacological Targeting of BET Bromodomain Proteins in Acute Myeloid Leukemia and Malignant Lymphomas: From Molecular Characterization to Clinical Applications"

_cancers, 2019, doi:10.3390/cancers11101483_

Round 1
Reviewer 1 Report
The Authors provide a nice overview of BET inhibitors. The manuscript is well written although it appears a little bit biased toward the Constellation compounds and does not fully cover the literature on the topic.
minor: page 2, line 93. I would skip "recent" since one of the refences is pretty old.
Author Response
Comment #1: “The Authors provide a nice overview of BET inhibitors. The manuscript is well written although it appears a little bit biased toward the Constellation compounds and does not fully cover the literature on the topic”.
Answer: we are grateful to the reviewer for his/her insightful comment. We have significantly modified the text by adding substantial information about a number of new compounds. Please see the section #4 of the manuscript (“Characterization and evaluation of BET inhibitors”) in which the changes have been highlighted in red to facilitate the reading.
Comment #2 “minor: page 2, line 93. I would skip "recent" since one of the refences is pretty old”.
Answer: we have made the corresponding modification in the new version of the manuscript.
Reviewer 2 Report
In this review article, Reyes-Garau et al. describes the effect of inhibition of BET proteins in hematological cancers. This paper is well organized and covers a wide-range of topics relevant to cancers and therapy.
This is not the first review regarding BET proteins in hematological malignancies since Abedin and colleagues pubblished a review on the same topic, in 2016 (DOI: 10.2147/ott.s100515). Nevertheless, the present article has a completely different structure. Moreover, since new scientific discoveries are made everyday, I think that if authors update bibliography as much as possible, this review could be of great interest for the readers of Cancers.
Broad comments
Following comments are mean to improve the paper and make it more attractive and comprehensible. For authors' convenience, remarks are divided according to the respective chapters.
Chapter 1: “ereasers” are stated as “sirtuins” while “histone deacetylases” are defined as “writers”. I disagree with this definition because HDAC could not be considered “writers” as HAT (histone acetyltransferase), they are formally defined as the “ereasers” of the histone code (i.e. DOI:10.1161/CIRCRESAHA.116.303630, DOI: 10.1101/cshperspect.a018713). Moreover, sirtuins represent only a subgroup (class III) of the HDAC family, thus defining them as the “ereasers” is misleading.
Chapter 2: description of BRD2 and BRD4 functions is good, but from line 93 onward authors describe the non-transcriptional functions of BRD4 focusing on a tumor context. In my opinion, this chapter suould remain general as the title suggest and the oncological implications should be addressed in the next section. Thus, I suggest to move below reference 30 and 31 and the relative sentences. Also reference 28 concern oncogenes, in particular Myc, but the relative sentence is not specific. Please do not generalize concepts or provide examples of other transcription factors.Moreover, I suggest to modify Figure 1 which actually describes a mechanism of transcription activation that does not only concern oncogenes.Finally, the non-transcriptional role of BRD4 is wider than the examples provided here, authors may want to broaden the discussion a little bit (these papers may help: DOI:10.1189/jlb.2ri0616-250r, DOI:10.1186/s12943-018-0915-9).
Chapter 3: examples are relevant and clearly explained. I suggest to move the sentence in lines 144 to 146 (or to add a similar sentence) at the end of this chapter to clealry recapitulate why BETi could be usefull in the treatment of hematological cancers. As a general consideration about the role of BET proteins in hematological malignancies, I would better explain its relationship with Myc. This topic is mentioned several times along the manuscript but readers would probably take advantage from an explicit and clear explanation of the control of BRD4 on Myc transcription, particularly beacuse Myc is a foundamental oncogene in several hematological tumors. A brief exlapnation of this relationship would make all the discussion clearer even for readers with less experience in this field.
Chapter 4: all BETi are presented clerarly and examples are appropriate. Since the review concern hematological cancers, I suggest to focus this chapter only on this setting. Therefore I would remove references to other tumors (i.e. USC, reference 65) or pathology (atherosclerosis, reference 72) and add to the title “hematological cancers” or similar. Maybe some other pubblication about mono and combination therapies would be a nice addition here (i.e.: DOI:10.1093/annonc/mdx157,DOI:10.18632/oncotarget.10983, DOI:10.1038/s41598-018-30509-3, DOI:10.3390/cancers11030304 ). This chapter is quite long, I suggest authors to split it in two maybe by separating monotherapy and combination, for ease of reading. I agree with the authors that a figure is very helpful to attract readers and discuss the concepts in a comprehensible manner, but actually I don't think Figure 2 is precise enough. TME is a little bit crowded and probably not essential in this context. Connections between citosolic pathways, BET proteins effects on chromatin and BETi are not very clear. Moreover, Figure 2 focus mainly on the CXCR4/CCL12 – Myc axis which is not (as described in text) the only mechanism. I think that a more precise and comprehensive figure (if necessary, more panels but with fewer components) would help readers to better understand all the pathways described.
Chapter 5: discussing the use of BETi to treat patients is the obvious continuation of the previous chapter but I expected a more substantial dissertation. I suggest authors to enrich this section, for example by further discuss clinical trials, at least those with results available (i.e. NCT02698189, NCT02431260) and by moving in this chapter the references about clinical trials with BETi present in chapter 4.
Chapter 6: mechanism of resistence to drugs is a very important topic and certainly it could not be restricted to a specific tumor type such as hematological malignancies. Nevertheless, this section discuss resistance to BETi depending on Myc-related mechanisms. I guess that authors choose this focus because Myc is one of the major oncogene driving hematological cancers but it is not well specified. As for chapter 3, I suggest to clarify the role of Myc so that readers can better understand why the mechanisms of resistence described here are all Myc-related. Otherwise, authors may want to broaden the framework of the mechanisms taken into consideration (suggestions, non-exhaustive list: DOI:10.1016/j.celrep.2018.02.011, DOI: 10.1186/s13046-018-0888-y, DOI:10.1016/j.molcel.2018.06.036)
Chapter 7: I really appreciated this section on new generation of inhibitors.
Specific comments
Minor comments mainly concern English errors, I suggest a proofreading by a native speaker.
There are many sentences too long and/or unclear. The following list is incomplete and intended only as an example: lines 18 to 20; lines 160 to 164; lines 356 to 361.
Other minor tips:
- in lines 263 to 267 is not explained in which cellular context “...additive effects were observed...”, sentence is vague
- check the word "through" throughout the manuscript, there are several typo (i.e. line 31)
- some expressions seem strange and a bit out of context, i.e. line 63 “signed up” could be substituted with “bind” or “recluted”; line 86 “keep up” could be substituted with “maintain”; line 285 “bench-to-the-bedside” is better known as “bench-to-bedside”
- in some sentences BET inhibitor is written “BETi's” even when possessive case is not required (i.e. line 334), please check throughout the manuscript
- in line 231 “cell lines” is repeated two times
- in line 354 “other” is probably a typo, perhaps authors meant “order”
Author Response
Comment #1: “Chapter 1: “ereasers” are stated as “sirtuins” while “histone deacetylases” are defined as “writers”. I disagree with this definition because HDAC could not be considered “writers” as HAT (histone acetyltransferase), they are formally defined as the “ereasers” of the histone code (i.e. DOI:10.1161/CIRCRESAHA.116.303630, DOI: 10.1101/cshperspect.a018713). Moreover, sirtuins represent only a subgroup (class III) of the HDAC family, thus defining them as the “ereasers” is misleading.”
Answer: we sincerely thank the reviewer for his/her insightful comment. The text was corrected in the new version of the manuscript (please see lines 38-39).
Comment #2: “Chapter 2: description of BRD2 and BRD4 functions is good, but from line 93 onward authors describe the non-transcriptional functions of BRD4 focusing on a tumor context. In my opinion, this chapter should remain general as the title suggest and the oncological implications should be addressed in the next section. Thus, I suggest to move below reference 30 and 31 and the relative sentences. Also reference 28 concern oncogenes, in particular Myc, but the relative sentence is not specific. Please do not generalize concepts or provide examples of other transcription factors. Moreover, I suggest to modify Figure 1 which actually describes a mechanism of transcription activation that does not only concern oncogenes.Finally, the non-transcriptional role of BRD4 is wider than the examples provided here, authors may want to broaden the discussion a little bit (these papers may help: DOI:10.1189/jlb.2ri0616-250r, DOI:10.1186/s12943-018-0915-9)”.….
Answer: Following reviewer’s comment, we removed the oncological information from this chapter, keeping the text focused in the general function of BET family. Also, the reference #28 was replaced
By the following one: “Donati, B.; Lorenzini, E.; Ciarrocchi, A. BRD4 and Cancer: Going beyond transcriptional regulation. Mol. Cancer 2018”, and the relative sentence was changed (highlighted in red).
The Figure 1 was modified to fulfill reviewer’s query. The new version now presents a “Schematic representation of BRD4 transcriptional activation”.
Lastly, following the reviewer suggestion, we broad the discussion about BRD4 a bit more (highlighted in red).
Comment #3: “Chapter 3: examples are relevant and clearly explained. I suggest to move the sentence in lines 144 to 146 (or to add a similar sentence) at the end of this chapter to clealry recapitulate why BETi could be usefull in the treatment of hematological cancers. As a general consideration about the role of BET proteins in hematological malignancies, I would better explain its relationship with Myc. This topic is mentioned several times along the manuscript but readers would probably take advantage from an explicit and clear explanation of the control of BRD4 on Myc transcription, particularly beacuse Myc is a foundamental oncogene in several hematological tumors. A brief exlapnation of this relationship would make all the discussion clearer even for readers with less experience in this field.”
Answer: According to reviewer’s suggestion, lines 144 to 146 were moved at the end of the chapter. Additionally, a short text describing the role of MYC in hematological cancers has been added (Highlighted in red).
Comment #4: “Chapter 4: all BETi are presented clerarly and examples are appropriate. Since the review concern hematological cancers, I suggest to focus this chapter only on this setting. Therefore I would remove references to other tumors (i.e. USC, reference 65) or pathology (atherosclerosis, reference 72) and add to the title “hematological cancers” or similar. Maybe some other pubblication about mono and combination therapies would be a nice addition here (i.e.: DOI:10.1093/annonc/mdx157,DOI:10.18632/oncotarget.10983, DOI:10.1038/s41598-018-30509-3, DOI:10.3390/cancers11030304 ). This chapter is quite long, I suggest authors to split it in two maybe by separating monotherapy and combination, for ease of reading. I agree with the authors that a figure is very helpful to attract readers and discuss the concepts in a comprehensible manner, but actually I don't think Figure 2 is precise enough. TME is a little bit crowded and probably not essential in this context. Connections between citosolic pathways, BET proteins effects on chromatin and BETi are not very clear. Moreover, Figure 2 focus mainly on the CXCR4/CCL12 – Myc axis which is not (as described in text) the only mechanism. I think that a more precise and comprehensive figure (if necessary, more panels but with fewer components) would help readers to better understand all the pathways described.”
Answer: We acknowledge the comment of the reviewer. Following his/her recommendation, chapter 4 was rewritten and now focuses exclusively on hematological cancers (See in red). In addition, this chapter was divided in two sections: 4.1 “Preclinical activity of BETis as monotherapies in hematological malignancies” and 4.2 “Evaluation of BETis as part of combinatorial therapies in preclinical settings”, and additional references were added to fully address single agent treatments and combination therapies (changes are highlighted in red).
Figure 2 was also modified to fully cover the mechanism of action of BETis in the clinics.
Comment #5: “Chapter 5: discussing the use of BETi to treat patients is the obvious continuation of the previous chapter but I expected a more substantial dissertation. I suggest authors to enrich this section, for example by further discuss clinical trials, at least those with results available (i.e. NCT02698189, NCT02431260) and by moving in this chapter the references about clinical trials with BETi present in chapter 4.”
Answer: We significantly improved the chapter 5 thanks to the suggestions of the reviewer (Please see the changes highlighted in red).
Comment #6: “Chapter 6: mechanism of resistence to drugs is a very important topic and certainly it could not be restricted to a specific tumor type such as hematological malignancies. Nevertheless, this section discuss resistance to BETi depending on Myc-related mechanisms. I guess that authors choose this focus because Myc is one of the major oncogene driving hematological cancers but it is not well specified. As for chapter 3, I suggest to clarify the role of Myc so that readers can better understand why the mechanisms of resistence described here are all Myc-related. Otherwise, authors may want to broaden the framework of the mechanisms taken into consideration (suggestions, non-exhaustive list: DOI:10.1016/j.celrep.2018.02.011, DOI: 10.1186/s13046-018-0888-y, DOI:10.1016/j.molcel.2018.06.036)”
Answer: Although we agree with the reviewer that resistance is an essential issue in the treatment of cancer by BET inhibitors in general and not only in hematological malignancies, we decided to report the information available in acute leukemia and malignant lymphoma in accordance with the general focus of the review. We believe that this way, we keep the consistency across the text. We also clarify the role of MYC in this context (please see in red).
Comment #7: “Minor comments mainly concern English errors, I suggest a proofreading by a native speaker”.
Answer: We made a great effort to improve the quality of the English as suggested by the Reviewer.
Comment #8: “There are many sentences too long and/or unclear. The following list is incomplete and intended only as an example: lines 18 to 20; lines 160 to 164; lines 356 to 361”.
Answer: The text was improved and the sentences were shortened according to reviewer’s suggestion.
Comment #9: “ Other minor tips: - in lines 263 to 267 is not explained in which cellular context “...additive effects were observed...”, sentence is vague
Answer: The text was improved following the suggestion of the reviewer.
Comment #10: “- check the word "through" throughout the manuscript, there are several typo (i.e. line 31)”
Answer: We thank the reviewer for his/her comment. We have carefully checked the typos in the present version of the manuscript.
Comment #11: “- - some expressions seem strange and a bit out of context, i.e. line 63 “signed up” could be substituted with “bind” or “recluted”; line 86 “keep up” could be substituted with “maintain”; line 285 “bench-to-the-bedside” is better known as “bench-to-bedside”
Answer: We thank the reviewer for highlighting these errors, that have been corrected in the present version of the manuscript.
Comment #12: “- - - in some sentences BET inhibitor is written “BETi's” even when possessive case is not required (i.e. line 334), please check throughout the manuscript”
Answer: We thank the reviewer for highlighting these errors, that have been corrected in the present version of the manuscript.
Comment #13: “- - in line 231 “cell lines” is repeated two times” - in line 354 “other” is probably a typo, perhaps authors meant “order”
Answer: We thank the reviewer for highlighting these mistakes, that have been corrected in the present version of the manuscript.
Reviewer 3 Report
The authors review and discuss the possibility of pharmacological targeting of the BET bromodomains in hematological cancers. They have a focus on the molecular interactions and the cellular functions of the bromodomains, and they also give an overview of the various inhibitors that have been developed. This therapeutic strategy is regarded as promising, and the article is definitely within the scope of the journal.
Major comment:
1. A focus of the reviewers is on the molecular and cellular functions of the BET bromodomains and less on the hematological malignancies. I would like the authors to include a much more in depth review and discussion of bromodomain functions and inhibitor effects especially in acute myeloid leukemia and malignant lymphomas where quite many studies have been published. Thus, I would like to see a much more detailed review of effects especially in these two malignancies. It should also be reflected in the title that this is a combined review including both a detailed presentation of the molecular/cellular functions in general and with an additional focus on the two hematological malignancies acute myeloid leukemia/lymphoma (possibly also a comparison of the effects in these two malignancies?).
2. The legends to the figures have to be improved and extended. I would also like to see one or two additional figures that would be helpful for the reader to understand the molecular interactions and intracellular functions. The authors should also organize their article into smaller sections with separate headings (one of the sections is two and a half page long). The lower part of figure 2 is rather confusing, and this figure should be improved.
3. The authors should consider to present results from clinical studies in Tables; it will then be easier for the reader to compare the results from different studies. There should be a focus both on the efficiency and the toxicity.
Minor comments:
1. Grammar/spelling/language need to be carefully controlled.
2. What about combination with intensive chemotherapy or postremission/posttransplant maintenance therapy in AML? More detailed comments should be made, e.g. for AML.
Author Response
Comment #1: “1. A focus of the reviewers is on the molecular and cellular functions of the BET bromodomains and less on the hematological malignancies. I would like the authors to include a much more in depth review and discussion of bromodomain functions and inhibitor effects especially in acute myeloid leukemia and malignant lymphomas where quite many studies have been published. Thus, I would like to see a much more detailed review of effects especially in these two malignancies. It should also be reflected in the title that this is a combined review including both a detailed presentation of the molecular/cellular functions in general and with an additional focus on the two hematological malignancies acute myeloid leukemia/lymphoma (possibly also a comparison of the effects in these two malignancies?)”
Answer: we sincerely thank the reviewer for his/her insightful comment. In the new version of the manuscript we have included a depth review and discussion of BET functions and inhibitory effects especially in AML and malignant lymphomas (please see the changes highlighted in red in the text). Following reviewer’s suggestion, we also changed the title to “Pharmacological targeting of BET bromodomain proteins in acute myeloid leukemia and malignant lymphomas: from molecular characterization to clinical applications”.
Comment #2: “2. The legends to the figures have to be improved and extended. I would also like to see one or two additional figures that would be helpful for the reader to understand the molecular interactions and intracellular functions. The authors should also organize their article into smaller sections with separate headings (one of the sections is two and a half page long). The lower part of figure 2 is rather confusing, and this figure should be improved.”
Answer: according to reviewer’s queries, figures 1 and 2 were modified to facilitate the understanding of the molecular interactions and intracellular functions related to bromodomain proteins. Also, chapter 4 was divided in two sections: 4.1 “Preclinical activity of BETis as monotherapies in hematological malignancies” and 4.2 “Evaluation of BETis as part of combinatorial therapies in preclinical settings”.
Comment #3: “ 3. The authors should consider to present results from clinical studies in Tables; it will then be easier for the reader to compare the results from different studies. There should be a focus both on the efficiency and the toxicity.”
Answer: we gratefully acknowledge the reviewer for his/her comment. The corrected version of Table 1 now includes the information (when available) of efficiency and toxicity reported by the promoters of the clinical trials (changes have been highlighted in red).
Comment #4: “Minor comments: 1. Grammar/spelling/language need to be carefully controlled”.
Answer: The English was substantially improved as suggested by the reviewer. Sentences were shortened and several typos were corrected.
Comment #5: “. What about combination with intensive chemotherapy or postremission/posttransplant maintenance therapy in AML? More detailed comments should be made, e.g. for AML.”
Answer: we agree with the reviewer that combination of intensive chemotherapy with epigenetic dugs is of particular interest, especially in AML patients. However, although several clinical studies have reported the benefits of associating some epidrugs like HDAC inhibitors with various forms of conventional chemotherapy such as nucleoside analogues (cytarabine, fludarabine), anthracyclines, and topoisomerase inhibitors (Ther Adv Hematol 2019; 10: 2040620718816698), as far as we know no information is available about the use of BET inhibitors in these settings. Therefore, we decided to not discuss this strategy in the present review.
Reviewer 4 Report
The authors summarize the development and efficacy of BET inhibitors in hematological cancers.
Overall, the review is well written but would definitely benefit from some clarity. The section 4 is much too long and not easy to read and is at times redundant. It can be substantially shortened in terms of which BET inhibitors to highlight more in depth in preclinical results.
In section 3, the authors refer to MLL rearranged AML as a frequent occurrence. This gene has been renamed KMT2A and this should be changed accordingly. Actually, KMT2A rearranged leukemias are rare in adults (5% of AML) and more common in children (22% of AML). This statement should be corrected.
I am missing a short description of the most common adverse effects associated with these inhibitors in clinical trials. The section on the clinical trials is very vague.
Author Response
Comment #1: “Overall, the review is well written but would definitely benefit from some clarity. The section 4 is much too long and not easy to read and is at times redundant. It can be substantially shortened in terms of which BET inhibitors to highlight more in depth in preclinical results”.
Answer: thanks to this insightful comment. To facilitate the reading, chapter 4 has been divided in two sections: 4.1 “Preclinical activity of BETis as monotherapies in hematological malignancies” and 4.2 “Evaluation of BETis as part of combinatorial therapies in preclinical settings”. Also, sentences were shortened and several typos were corrected.
Comment #2: “In section 3, the authors refer to MLL rearranged AML as a frequent occurrence. This gene has been renamed KMT2A and this should be changed accordingly. Actually, KMT2A rearranged leukemias are rare in adults (5% of AML) and more common in children (22% of AML). This statement should be corrected.”
Answer: we gratefully acknowledge the reviewer for his/her comment. Text was updated according to reviewer’s suggestion (text highlighted in red, lines 185-193).
Comment #3: “In I am missing a short description of the most common adverse effects associated with these inhibitors in clinical trials. The section on the clinical trials is very vague.”.
Answer: we gratefully acknowledge the reviewer for his/her comment. The corrected version of Table 1 now includes the information (when available) of efficiency and toxicity reported by the promoters of the clinical trials (changes have been highlighted in red).
Round 2
Reviewer 2 Report
I think that Authors have addressed all the major comments. I appreciate their efforts.
Author Response
We gratefully thank the reviewer for his/her comment and support during the revision process.
Reviewer 3 Report
The authors have definitely improved their review by including several new sectons/chapters. However, in my opinion the review can still be improved to make the text easier to read.
Introduction: Several sections are very long and should be divided into several shorter sections. For example. the long section lines 96-126 can easily be divided into shorter sections. The authors should go through the whole text and consider whether other sections can be divided into shorter sections. They should also consider the order of individual sections in certain parts of the manuscript.
I suggest that instead of BETis and HDACis they write BET inhibitors and HDAC inhibitors. I would also suggest that they consider to use experimental instead of preclinical studies.
The question of toxicity is very important. I would suggest that they only present ongoing studies in Table 1 and that they present observations from reported studies in a separate table. This should be done more in detail and has to include age, number of patients, clinical characteristics and more detailed descriptions of responses and toxicity. The toxicity is a very important issue and should in my opinion be described and discussed in a separate section. For each type of toxicity they should describe how frequent and how serious. With regard to NFkB, what about immunosuppression/immunotoxicity.
Author Response
Answers to Reviewer #3
Comment #1: “Several sections are very long and should be divided into several shorter sections. For example. the long section lines 96-126 can easily be divided into shorter sections. The authors should go through the whole text and consider whether other sections can be divided into shorter sections. They should also consider the order of individual sections in certain parts of the manuscript.”.
Answer: Following referee’s comment, we have included an extra section within the chapter 2 (2.1 BRD4: Biological roles and molecular mechanisms of action over chromatin structure). Since BRD4 is the best studied member of the BET family of proteins, a large amount of information regarding this protein in presented in the article. Thus, we have agreed in including it in a separate section within the chapter 2.
In addition, chapter 3 was split in two parts. In the first one “3.1 MYC-driven mechanisms of oncogenesis regulated by BET proteins”, we point out the molecular mechanisms of BET-mediated expression for the MYC oncogene in particular. In the second subchapter “3.2 BET-mediated expression regulation of other oncogenes”, we focus in the BET-mediated transcriptional regulation of additional oncogenes.
Finally, the English has been extensively revised and the text has been modified (highlighted in red in the enclosed document) to make the manuscript more readable.
Comment #2: “I suggest that instead of BETis and HDACis they write BET inhibitors and HDAC inhibitors. I would also suggest that they consider to use experimental instead of preclinical studies”.
Answer: we are grateful to the referee for his/her comment. Accordingly, we have incorporated these modifications along the whole text.
Comment #3: “The question of toxicity is very important. I would suggest that they only present ongoing studies in Table 1 and that they present observations from reported studies in a separate table. This should be done more in detail and has to include age, number of patients, clinical characteristics and more detailed descriptions of responses and toxicity. The toxicity is a very important issue and should in my opinion be described and discussed in a separate section. For each type of toxicity they should describe how frequent and how serious. With regard to NFkB, what about immunosuppression/immunotoxicity.”
Answer: to comply with referee’s query, we have restricted the information contained in Table 1 to ongoing clinical trials using BET inhibitors and we have created the new Table 2, which gathers the clinical and biological characteristics of the patients together with detailed side effects for each BETi tested in the clinics. Additionally, chapter 5 has been extended (lines 393-399) to address more in detail the toxicity reported so far in these different clinical trials.
Regarding the impact of BETi treatment on NFkB-related immunosuppression/immunotoxicity, we have been unable to find relevant published data in the leukemia/lymphoma field. Besides the molecular mechanism by which BRD4 activates NFkB (detailed in lines 99-105), some reports centered on solid tumors have suggested that co-inhibition of BET proteins and NFkB can be of potential interest (see for example PMID:29472532). However, as the present review is focused on hematological malignancies, we discarded to mention these studies here.
Reviewer 4 Report
no further comments
Author Response
We gratefully thank the reviewer for his/her comment. The English and the phrasing have been extensively revised and the text has been modified (highlighted in red in the enclosed document) to make the manuscript more readable.